# Training Autoencoders by Alternating Minimization

## Abstract

We present DANTE, a novel method for training neural networks, in particular autoencoders, using the alternating minimization principle. DANTE provides a distinct perspective in lieu of traditional gradient-based backpropagation techniques commonly used to train deep networks. It utilizes an adaptation of quasi-convex optimization techniques to cast autoencoder training as a bi-quasi-convex optimization problem. We show that for autoencoder configurations with both differentiable (e.g. sigmoid) and non-differentiable (e.g. ReLU) activation functions, we can perform the alternations very effectively. DANTE effortlessly extends to networks with multiple hidden layers and varying network configurations. In experiments on standard datasets, autoencoders trained using the proposed method were found to be very promising and competitive to traditional backpropagation techniques, both in terms of quality of solution, as well as training speed.

## 1 Introduction

For much of the recent march of deep learning, gradient-based backpropagation methods, e.g. Stochastic Gradient Descent (SGD) and its variants, have been the mainstay of practitioners. The use of these methods, especially on vast amounts of data, has led to unprecedented progress in several areas of artificial intelligence. On one hand, the intense focus on these techniques has led to an intimate understanding of hardware requirements and code optimizations needed to execute these routines on large datasets in a scalable manner. Today, myriad off-the-shelf and highly optimized packages exist that can churn reasonably large datasets on GPU architectures with relatively mild human involvement and little bootstrap effort.

However, this surge of success of backpropagation-based methods in recent years has somewhat overshadowed the need to continue to look for options beyond backprogagation to train deep networks. Despite several advancements in deep learning with respect to novel architectures such as encoder-decoder networks and generative adversarial models, the reliance on backpropagation methods remains. While reinforcement learning methods are becoming increasingly popular, their scope is limited to a particular family of settings such as agent-based systems or reward-based learning. Recent efforts have studied the limitations of SGD-based backpropagation, including parallelization of SGD-based techniques that are inherently serial (Taylor et al. (2016)); vanishing gradients, especially for certain activation functions (Hochreiter & Schmidhuber (1997)); convergence of stochastic techniques to local optima (Anandkumar & Ge (2016)); and many more. For a well-referenced recent critique of gradient-based methods, we point the reader to Taylor et al. (2016).

From another perspective, there has been marked progress in recent years in the area of non-convex optimization (beyond deep learning), which has resulted in scalable methods such as iterated hard thresholding (Blumensath & Davies (2009)) and alternating minimization (Jain et al. (2013)) as methods of choice for solving large-scale sparse recovery, matrix completion, and tensor factorization tasks. Several of these methods not only scale well to large problems, but also offer provably accurate solutions. In this work, we investigate a non-backpropagation strategy to train neural networks, leveraging recent advances in quasi-convex optimization. Our method is called DANTE (Deep AlterNations for Training autoEncoders), and it offers an alternating minimization-based technique for training neural networks - in particular, autoencoders.

DANTE is based on a simple but useful observation that the problem of training a single hidden-layer autoencoder can be cast as a bi-quasiconvex optimization problem (described in Section 3.1). This

observation allows us to use an alternating optimization strategy to train the autoencoder, where each step involves relatively simple quasi-convex problems. DANTE then uses efficient solvers for quasi-convex problems including normalized gradient descent (Nesterov (1984)) and stochastic normalized gradient descent (Hazan et al. (2015)) to train autoencoder networks. The key contributions of this work are summarized below:

- We show that viewing each layer of a neural network as applying an ensemble of generalized linear transformations, allows the problem of training the network to be cast as a bi-quasi-convex optimization problem (exact statement later).

- We exploit this intuition by employing an alternating minimization strategy, DANTE, that reduces the problem of training the layers to quasi-convex optimization problems.

- We utilize the state-of-the-art Stochastic Normalized Gradient Descent (SNGD) technique (Hazan et al. (2015)) for quasi-convex optimization to provide an efficient implementation of DANTE for networks with sigmoidal activation functions. However, a limitation of SNGD is its inability to handle non-differentiable link functions such as the ReLU.

- To overcome this limitation, we introduce the *generalized ReLU*, a variant of the popular ReLU activation function and show how SNGD may be applied with the generalized ReLU function. This presents an augmentation in the state-of-the-art in quasi-convex optimization and may be of independent interest. This allows DANTE to train AEs with both differentiable and non-differentiable activation functions, including ReLUs and sigmoid.

- We show that SNGD offers provably more rapid convergence with the generalized ReLU function than it does even for the sigmoidal activation. This is corroborated in experiments as well. A key advantage of our approach is that these theoretical results can be used to set learning rates and batch sizes without finetuning/cross-validation.

- We also show DANTE can be easily extended to train deep AEs with multiple hidden layers.

- We empirically validate DANTE with both the generalized ReLU and sigmoid activations and establish that DANTE provides competitive test errors, reconstructions and classification performance (with the learned representations), when compared to an identical network trained using standard mini-batch SGD-based backpropagation.

## 2 RELATED WORK

Backpropagation-based techniques date back to the early days of neural network research (Rumelhart et al. (1986); Chauvin & Rumelhart (1995)) but remain to this day, the most commonly used methods for training a variety of neural networks including multi-layer perceptrons, convolutional neural networks, autoencoders, recurrent networks and the like. Recent years have seen the development of other methods, predominantly based on least-squares approaches, used to train neural networks. Carreira-Perpinan and Wang (Carreira-Perpinan & Wang (2014)) proposed a least-squares based method to train a neural network. In particular, they introduced the Method of Auxiliary Constraints (MAC), and used quadratic penalties to enforce equality constraints. Patel *et al.* (Patel et al. (2015)) proposed an Expectation-Maximization (EM) approach derived from a hierarchical generative model called the Deep Rendering Model (DRM), and also used least-squared parameter updates in each of the EM steps. They showed that forward propagation in a convolutional neural network was equivalent to the inference on their DRM. Unfortunately, neither of these methods has publicly available implementations or published training results to compare against.

More recently, Taylor *et al.* proposed a method to train neural networks using the Alternating Direction Method of Multipliers (ADMM) and Bregman iterations (Taylor et al. (2016)). The focus of this method, however, was on scaling the training of neural networks to a distributed setting on multiple cores across a computing cluster. Jaderberg also proposed the idea of 'synthetic gradients' in Jaderberg et al. (2016). While this approach is interesting, this work is more focused towards a more efficient way to carry out gradient-based parameter updates in a neural network.

In our work, we focus on an entirely new approach to training neural networks – in particular, autoencoders – using alternating optimization, quasi-convexity and SNGD, and show that this approach shows promising results on the a range of datasets. Although alternating minimization has found much appeal in areas such as matrix factorization (Jain et al. (2013)), to the best of our knowledge, this is the first such effort in using alternating principles to train neural networks with related performance guarantees.

## 3 DANTE: DEEP ALTERNATIONS FOR TRAINING AUTOENCODERS

In this section, we will first set notation and establish the problem setting, then present details of the DANTE method, including the SNGD algorithm. For sake of simplicity, we consider networks with just a single hidden layer. We then offer some theoretical insight intro DANTE's inner workings, which also allow us to arrive at the generalized ReLU activation function, and finally describe how DANTE can be extended to deep networks with multiple hidden layers.

### 3.1 PROBLEM FORMULATION

Consider a neural network with $L$ layers. Each layer $l \in \{1, 2, \ldots, L\}$ has $n_l$ nodes and is characterized by a linear operator $W_l \in \mathbb{R}^{n_{l-1} \times n_l}$ and a non-linear activation function $\phi_l : \mathbb{R}^{n_l} \to \mathbb{R}^{n_l}$. The activations generated by the layer $l$ are denoted by $\mathbf{a}_l \in \mathbb{R}^{n_l}$. We denote by $\mathbf{a}_0$, the input activations and $n_0$ to be the number of input activations i.e. $\mathbf{a}_0 \in \mathbb{R}^{n_0}$. Each layer uses activations being fed into it to compute its own activations as $\mathbf{a}_l = \phi_l \langle W_l, \mathbf{a}_{l-1} \rangle \in \mathbb{R}^{n_l}$, where $\phi \langle ., . \rangle$ denotes $\phi(\langle ., . \rangle)$ for simplicity of notation. A multi-layer neural network is formed by nesting such layers to form a composite function $f$ given as follows:

$$f(\mathbf{W}; \mathbf{x}) = \phi_L \langle W_L, \phi_{L-1} \langle W_{L-1}, \cdots, \phi_1 \langle W_1, \mathbf{x} \rangle \rangle \rangle \tag{1}$$

where $\mathbf{W} = \{W_l\}$ is the collection of all the weights through the network, and $\mathbf{x} = \mathbf{a}_0$ contains the input activations for each training sample.

Given $m$ data samples $\{(\mathbf{x_i}, y_i)\}_{i=1}^m$ from some distribution $\mathcal{D}$, the network is trained by tuning the weights $\mathbf{W}$ to minimize a given loss function, $J$:

$$\min_{\mathbf{W}} \ \mathbb{E}_{(\mathbf{x},y) \sim \mathcal{D}}[J(f(\mathbf{W}; \mathbf{x}), y)] \tag{2}$$

Note that a multi-layer autoencoder is trained similarly, but with the loss function modified as below:

$$\min_{\mathbf{W}} \ \mathbb{E}_{\mathbf{x} \sim \mathcal{D}}[J(f(\mathbf{W}; \mathbf{x}), \mathbf{x})] \tag{3}$$

For purpose of simplicity and convenience, we first consider the case of a single-layer autoencoder, represented as $f(\mathbf{W}; \mathbf{x}) = \phi_2 \langle W_2, \phi_1 \langle W_1, \mathbf{x} \rangle \rangle$ to describe our methodology. We describe in a later section on how this idea can be extended to deep multi-layer autoencoders. (Note that our definition of a single-layer autoencoder is equivalent to a two-layer neural network in a classification setting, by nature of the autoencoder.)

A common loss function used to train autoencoders is the squared loss function which, in our simplified setting, yields the following objective.

$$\min_{\mathbf{W}} \ \mathbb{E}_{\mathbf{x} \sim \mathcal{D}} \|f(\mathbf{W}; \mathbf{x}) - \mathbf{x}\|_2^2 \tag{4}$$

Now denote

$$\|f(\mathbf{W}; \mathbf{x}) - \mathbf{x}\|_2^2 = \|\phi_2 \langle W_2, \phi_1 \langle W_1, \mathbf{x} \rangle \rangle - \mathbf{x}\|_2^2 \tag{5}$$

An important observation here is that if we fix $W_1$, then Eqn (5) turns into a set of Generalized Linear Model problems with $\phi_2$ as the activation function, i.e.

$$\min_{W_2} \ \mathbb{E}_{\mathbf{x} \sim \mathcal{D}} \|\phi_2 \langle W_2, \mathbf{z} \rangle - \mathbf{x}\|_2^2$$

where $\mathbf{z} = \phi_1 \langle W_1, \mathbf{x} \rangle$. We exploit this observation in this work. In particular, we leverage a recent result by Hazan et al. (2015) that shows that GLMs with nice, differentiable link functions such as sigmoid (or even a combination of sigmoids such as $\phi_{W_2}(\cdot)$), satisfy a property the authors name *Strict Locally Quasi-Convexity* (SLQC), which allows techniques such as SNGD to solve the GLM problems effectively.

Similarly, fixing $W_2$ turns the problem into yet another SLQC problem, this time with $W_1$ as the parameter (note that $\phi_{W_2} \langle \cdot \rangle = \phi_2 \langle W_2, \phi_1 \langle \cdot \rangle \rangle$).

$$\min_{W_1} \ \mathbb{E}_{\mathbf{x} \sim \mathcal{D}} \|\phi_{W_2} \langle W_1, \mathbf{x} \rangle - \mathbf{x}\|_2^2.$$

---

**Algorithm 1:** Stochastic Normalized Gradient Descent (SNGD)

**Input** : Number of iterations $T$, training data $S = \{(\mathbf{x}_i, y_i)\}_{i=1}^m \in \mathbb{R}^d \times \mathbb{R}$, learning rate $\eta$,
minibatch size $b$, Initialization parameters $\mathbf{w}_0$

1  **for** $t = 1$ *to* $T$ **do**
2  |  Sample $\{(\mathbf{x}_i, y_i)\}_{i=1}^b \sim \mathsf{Uniform}(S)$  //Select a random mini-batch of training points
3  |  Let $f_t(\mathbf{w}) = \frac{1}{b} \sum_{i=1}^b (y_i - \phi\langle\mathbf{w}, \mathbf{x}_i\rangle)^2$
4  |  Let $\mathbf{g}_t = \nabla f_t(\mathbf{w}_t)$, and $\hat{\mathbf{g}}(t) = \frac{\mathbf{g}_t}{\|\mathbf{g}_t\|}$
5  |  $\mathbf{w}_{t+1} = \mathbf{w}_t - \eta \cdot \hat{\mathbf{g}}_t$
6  **end**
**Output** : Model given by $\mathbf{w}_T$

---

This is quite advantageous for us since it allows us to solve each sub-problem of the alternating setup efficiently. In a subsequent section, we will show that GLMs with non-differentiable activation – in particular, a generalized Rectified Linear Unit (ReLU) – can also satisfy the SLQC property, thus allowing us to extend the proposed alternating strategy, DANTE, to ReLU-based autoencoders too. We note that while we have developed this idea to train autoencoders in this work (since our approach relates closely to the greedy layer-wise training in autoencoders), DANTE can be used to train standard multi-layer neural networks too (discussed in Section 5).

### 3.2 METHODOLOGY

We begin our presentation of the proposed method by briefly reviewing the Stochastic Normalized Gradient Descent (SNGD) method, which is used to execute the inner steps of DANTE. We explain in the next subsection, the rationale behind the choice of SNGD as the optimizer. We stress that although DANTE does use stochastic gradient-style methods internally (such as the SNGD algorithm), the overall strategy adopted by DANTE is not a descent-based strategy, rather an alternating-minimization strategy.

**Stochastic Normalized Gradient Descent (SNGD):** Normalized Gradient Descent (NGD) is an adaptation of traditional Gradient Descent where the updates in each iteration are purely based on the direction of the gradients, while ignoring their magnitudes. This is achieved by normalizing the gradients. SNGD is the stochastic version of NGD, where weight updates are performed using individual (randomly chosen) training samples, instead of the complete set of samples. Mini-batch SNGD generalizes this by applying updates to the parameters at the end of every mini-batch of samples, as does mini-batch Stochastic Gradient Descent (SGD). In the remainder of this paper, we refer to mini-batch SNGD as SNGD itself, as is common for SGD. Algorithm 1 describes the SNGD methodology for a generic GLM problem.

**DANTE:** Given this background, Algorithm 2 outlines the proposed method, DANTE. Consider the autoencoder problem below for a single hidden layer network:

$$\min_{\mathbf{W}} f(W_1, W_2) = \mathbb{E}_{\mathbf{x}\sim\mathcal{D}}\|\phi_2\langle W_2, \phi_1\langle W_1, \mathbf{x}\rangle\rangle - \mathbf{x}\|_2^2$$

Upon fixing the parameters of the lower layer i.e. $W_1$, it is easy to see that we are left with a set of GLM problems:

$$\min_W \mathbb{E}_{\mathbf{x}\sim\mathcal{D}}\|\phi_2\langle W_2, \mathbf{z}\rangle - \mathbf{x}\|_2^2,$$

where $\mathbf{z} = \phi_1\langle W_1, \mathbf{x}\rangle$. DANTE solves this intermediate problem using SNGD steps by sampling several mini-batches of data points and performing updates as dictated by Algorithm 1. Similarly, fixing the parameters of the upper layer, i.e. $W_2$, we are left with another set of problems:

$$\min_W \mathbb{E}_{\mathbf{x}\sim\mathcal{D}}\|\phi_{W_2}\langle W_1, \mathbf{x}\rangle - \mathbf{x}\|_2^2,$$

where $\phi_{W_2}\langle\cdot\rangle = \phi_2\langle W_2, \phi_1\langle\cdot\rangle\rangle$. This is once again solved by mini-batch SNGD, as before.

---

**Algorithm 2:** DANTE: Deep AlterNations for Training autoEncoders

---

**Input** : Stopping threshold $\epsilon$, Number of iterations of alternating minimization $T_{AM}$, Number of
iterations for SNGD $T_{SNGD}$, initial values $W_1^0, W_2^0$, learning rate $\eta$, minibatch size $b$

1   $t := 1$

2   **while** $|f(W_1^t, W_2^t) - f(W_1^{t-1}, W_2^{t-1})| \geq \epsilon$ ***or*** $t < T_{AM}$ **do**

3     $W_2^t \leftarrow \underset{W}{\arg\min} \ \mathbb{E}_{\mathbf{x} \sim \mathcal{D}} \|\phi_2\langle W, \phi_1\langle W_1^{t-1}, \mathbf{x}\rangle\rangle - \mathbf{x}\|_2^2$         //use SNGD

4     $W_1^t \leftarrow \underset{W}{\arg\min} \ \mathbb{E}_{\mathbf{x} \sim \mathcal{D}} \|\phi_2\langle W_2^t, \phi_1\langle W, \mathbf{x}\rangle\rangle - \mathbf{x}\|_2^2$         //use SNGD

5     $t := t + 1$

6   **end**

   **Output** : $W_1^t, W_2^t$

---

### 3.3 RATIONALE

To describe the motivation for our alternating strategy in DANTE, we first define key terms and results that are essential to our work. We present the notion of a locally quasi-convex function (as introduced in Hazan et al. (2015)) and show that under certain realizability conditions, empirical objective functions induced by Generalized Linear Models (GLMs) are locally quasi-convex. We then introduce a new activation function, the generalized ReLU, and show that the GLM with the generalized ReLU also satisfies this property. We cite a result that shows that SNGD converges to the optimum solution provably for locally quasi-convex functions, and subsequently extend this result to the newly introduced activation function. We also generalize the definition of locally quasi-convex to functions on matrices, which allows us to relate these ideas to layers in neural networks.

**Definition 3.1** (*Local-Quasi-Convexity*). Let $\mathbf{x}, \mathbf{z} \in \mathbb{R}^d, \kappa, \epsilon > 0$ and let $f : \mathbb{R}^d \to \mathbb{R}$ be a differentiable function. Then $f$ is said to be $(\epsilon, \kappa, \mathbf{z})$-Strictly-Locally-Quasi-Convex (SLQC) in $\mathbf{x}$, if at least one of the following applies:

     1. $f(\mathbf{x}) - f(\mathbf{z}) \leq \epsilon$

     2. $\|\nabla f(\mathbf{x})\| > 0$, and $\forall \mathbf{y} \in \mathbb{B}(\mathbf{z}, \epsilon/\kappa), \langle \nabla f(\mathbf{x}), \mathbf{y} - \mathbf{x}\rangle \leq 0$

where $\mathbb{B}(\mathbf{z}, \epsilon/\kappa)$ refers to a ball centered at $\mathbf{z}$ with radius $\epsilon/\kappa$.

We generalize this definition to functions on matrices in Appendix A.3.

**Definition 3.2** (*Idealized and Noisy Generalized Linear Model (GLM)*). Given an (unknown) distribution $\mathcal{D}$ and an activation function $\phi : \mathbb{R} \to \mathbb{R}$, an idealized GLM is defined by the existence of a $\mathbf{w}^* \in \mathbb{R}^d$ such that $y_i = \phi\langle\mathbf{w}^*, \mathbf{x}_i\rangle \forall i \in [m]$ where $\mathbf{w}^*$ is the global minimizer of the error function:

$$\hat{err}(\mathbf{w}) = \frac{1}{m} \sum_{i=1}^{m} (y_i - \phi(\langle\mathbf{w}, \mathbf{x_i}\rangle))^2$$

Similarly, a noisy GLM is defined by the existence of a $\mathbf{w}^* \in \mathbb{R}^d$ such $\mathbb{E}_{(\mathbf{x},y)\sim\mathcal{D}}[y \,|\, \mathbf{x}] = \phi(\langle\mathbf{w}^*, \mathbf{x}\rangle)$, which is the global minimizer of the error function:

$$err(\mathbf{w}) = \mathbb{E}_{(\mathbf{x},y)\sim\mathcal{D}} (y_i - \phi(\langle\mathbf{w}, \mathbf{x_i}\rangle))^2$$

Without any loss in generality, we use $\mathbf{x}_i \in \mathbb{B}_d$, the unit $d$-dimensional ball.

(Hazan et al., 2015, Lemma 3.2) shows that if we draw $m \geq \Omega\left(\frac{\exp(2\|\mathbf{w}^*\|)}{\epsilon^2} \log \frac{1}{\delta}\right)$ samples of $\{(\mathbf{x}_i, y_i)\}_{i=1}^m$ from a GLM with the sigmoid activation function, then with probability at least $1 - \delta$, the empirical error function

$$f(\mathbf{w}) = \frac{1}{m} \sum_{i=1}^{m} (y_i - \phi\langle\mathbf{w}, \mathbf{x}_i\rangle)^2$$

is $(\epsilon, e^{\|\mathbf{w}^*\|_2}, \mathbf{w}^*)$-SLQC in $\mathbf{w}$. However, this result is restrictive, since its proof relies on properties of the sigmoid function, which are not satisfied by other popular activation functions such as the ReLU. We hence introduce a new generalized ReLU activation function to study the relevance of this result in a broader setting (which has more use in practice).

**Definition 3.3.** *(Generalized ReLU)* The generalized ReLU function $f : \mathbb{R} \to \mathbb{R}$, $0 < a < b$, $a, b \in \mathbb{R}$ is defined as:

$$f(x) = \begin{cases} ax & x \leq 0 \\ bx & x > 0 \end{cases}$$

This function is differentiable at every point except 0. Note that this definition subsumes variants of ReLU such as the leaky ReLU (Xu et al. (2015)). We define the function $g$ that provides a valid subgradient for the generalized ReLU at all $x$ to be:

$$g(x) = \begin{cases} a & x < 0 \\ b & x \geq 0 \end{cases}$$

While SLQC is originally defined for differentiable functions, we now show that with the above definition of the subgradient, the GLM with the generalized ReLU is also SLQC. This allows us to use the SNGD as an effective optimizer for DANTE to train autoencoders with different kinds of activation functions.

**Theorem 3.4.** *In the idealized GLM with generalized ReLU activation, assuming $||\mathbf{w}^*|| \leq W$, $\hat{err}(\mathbf{w})$ is $\left(\epsilon, \frac{2b^3 W}{a}, \mathbf{w}^*\right) - SLQC$ in $\mathbf{w}$ for all $\boldsymbol{w} \in \mathbb{B}_d(0, W)$.*

*Proof.* Consider $||\mathbf{w}|| \leq W$ such that $\hat{err}_m(\mathbf{w}) = \frac{1}{m} \sum_{i=1}^m (y_i - \phi\langle \mathbf{w}, \mathbf{x}_i \rangle)^2 \geq \epsilon$, where $m$ is the total number of samples. Also let $\mathbf{v}$ be a point $\epsilon/\kappa$-close to minima $\mathbf{w}^*$ with $\kappa = \frac{2b^3 W}{a}$. Let $g$ be the subgradient of the generalized ReLU activation and $G$ be the subgradient of $\hat{err}_m(\mathbf{w})$. (Note that as before, $g\langle ., . \rangle$ denotes $g(\langle ., . \rangle)$). Then:

$$\langle G(\mathbf{w}), \mathbf{w} - \mathbf{v} \rangle$$

$$= \frac{2}{m} \sum_{i=1}^m g\langle \mathbf{w}, \mathbf{x}_i \rangle \left( \phi\langle \mathbf{w}, \mathbf{x}_i \rangle - y_i \right) \langle \mathbf{x}_i, (\mathbf{w} - \mathbf{v}) \rangle$$

$$= \frac{2}{m} \sum_{i=1}^m g\langle \mathbf{w}, \mathbf{x}_i \rangle \left( \phi\langle \mathbf{w}, \mathbf{x}_i \rangle - \phi\langle \mathbf{w}^*, \mathbf{x}_i \rangle \right) \left[ \langle \mathbf{x}_i, \mathbf{w} - \mathbf{w}^* \rangle + \langle \mathbf{x}_i, \mathbf{w}^* - \mathbf{v} \rangle \right] \quad \text{(Step 1)}$$

$$\geq \frac{2}{m} \sum_{i=1}^m g\langle \mathbf{w}, \mathbf{x}_i \rangle \left[ b^{-1} \left( \phi\langle \mathbf{w}, \mathbf{x}_i \rangle - \phi\langle \mathbf{w}^*, \mathbf{x}_i \rangle \right)^2 + \left( \phi\langle \mathbf{w}, \mathbf{x}_i \rangle - \phi\langle \mathbf{w}^*, \mathbf{x}_i \rangle \right) \langle \mathbf{x}_i, \mathbf{w}^* - \mathbf{v} \rangle \right]$$

$$\text{(Step 2)}$$

$$\geq \frac{2}{m} \sum_{i=1}^m g\langle \mathbf{w}, \mathbf{x}_i \rangle \left( b^{-1} \left( \phi\langle \mathbf{w}, \mathbf{x}_i \rangle - \phi\langle \mathbf{w}^*, \mathbf{x}_i \rangle \right)^2 - |\phi\langle \mathbf{w}, \mathbf{x}_i \rangle - \phi\langle \mathbf{w}^*, \mathbf{x}_i \rangle| ||\mathbf{x}_i|| ||\mathbf{w}^* - \mathbf{v}|| \right.$$

$$\geq \frac{2}{m} \sum_{i=1}^m ab^{-1} \left( \phi\langle \mathbf{w}, \mathbf{x}_i \rangle - \phi\langle \mathbf{w}^*, \mathbf{x}_i \rangle \right)^2 - b|\phi\langle \mathbf{w}, \mathbf{x}_i \rangle - \phi\langle \mathbf{w}^*, \mathbf{x}_i \rangle| ||\mathbf{x}_i|| ||\mathbf{w}^* - \mathbf{v}|| \quad \text{(Step 3)}$$

$$\geq \frac{2}{m} \sum_{i=1}^m ab^{-1} \left( \phi\langle \mathbf{w}, \mathbf{x}_i \rangle - \phi\langle \mathbf{w}^*, \mathbf{x}_i \rangle \right)^2 - b^2 ||\langle \mathbf{w}, \mathbf{x}_i \rangle - \langle \mathbf{w}^*, \mathbf{x}_i \rangle || \frac{\epsilon}{\kappa} ||\mathbf{x}_i|| \quad \text{(Step 4)}$$

$$\geq 2ab^{-1}\epsilon - \frac{a\epsilon}{bW} ||\langle \mathbf{w}, \mathbf{x}_i \rangle - \langle \mathbf{w}^*, \mathbf{x}_i \rangle || ||\mathbf{x}_i|| \quad \text{(Step 5)}$$

$$\geq ab^{-1}\epsilon (2 - \frac{1}{W} ||\mathbf{w} - \mathbf{w}^*|| ||\mathbf{x}_i||^2)$$

$$\geq 0 \qquad \qquad \qquad \qquad \qquad \qquad \qquad \qquad \qquad \qquad \qquad \qquad \qquad \qquad \qquad \square$$

In the above proof, we first use the fact (in Step 1) that in the GLM, there is some $\mathbf{w}^*$ such that $\phi\langle \mathbf{w}^*, \mathbf{x}_i \rangle = y_i$. Then, we use the fact (in Steps 2 and 4) that the generalized ReLU function is $b$-Lipschitz, and the fact that the minimum value of the quasigradient of $g$ is $a$ (Step 3). Subsequently, in Step 5, we simply use the given bounds on the variables $\mathbf{x}_i, \mathbf{w}, \mathbf{w}^*$ due to the setup of the problem ($\mathbf{w} \in \mathbb{B}_d(0, W)$, and $\mathbf{x}_i \in \mathbb{B}_d$, the unit $d$-dimensional ball, as defined earlier in this section).

We also prove a similar result for the Noisy GLM below.

**Theorem 3.5.** *In the noisy GLM with generalized ReLU activation, assuming $||\mathbf{w}^*|| \leq W$, given $w \in B(0, W)$, then with probability $\geq 1 - \delta$ after $m \geq \frac{288b^4W^2}{a^2\epsilon^2}log(1/\delta)/\epsilon^2$ samples, $\hat{err}(\mathbf{w})$ is $\left(\epsilon, \frac{2b^3W}{a}, \mathbf{w}^*\right) - SLQC$ in $\mathbf{w}$.*

The proof for Theorem 3.5 is included in Appendix A.1.

We connect the above results with a result from Hazan et al. (2015) (stated below) which shows that SNGD provably converges to the optimum for SLQC functions, and hence, with very high probability, for empirical objective functions induced by noisy GLM instances too.

**Theorem 3.6** (Hazan et al. (2015))**.** *Let $\epsilon, \delta, G, M, \kappa > 0$, let $f : \mathbb{R}^d \to \mathbb{R}$ and $\mathbf{w}^* = \arg\min_{\mathbf{w}} f(\mathbf{w})$. Assume that for $b \geq b_0(\epsilon, \delta, T)$, with probability $\geq 1 - \delta$, $f_t$ defined in Algorithm 1 is $(\epsilon, \kappa, \mathbf{w}^*)$-SLQC $\forall \mathbf{w}$, and $|f_t| \leq M \forall t \in \{1, \cdots, T\}$. If we run SNGD with $T \geq \frac{\kappa^2 ||\mathbf{w}_1 - \mathbf{w}^*||^2}{\epsilon^2}$ and $\eta = \frac{\epsilon}{\kappa}$, and $b \geq \max\left\{\frac{M^2 log\left(\frac{4T}{\delta}\right)}{2\epsilon^2}, b_0(\epsilon, \delta, T)\right\}$, with probability $1 - 2\delta$, $f(\mathbf{w}) - f(\mathbf{w}^*) \leq 3\epsilon$.*

The results so far show that SNGD provides provable convergence for idealized and noisy GLM problems with both sigmoid and ReLU family of activation functions. We note that alternate activation functions such as $\tanh$ (which is simply a rescaled sigmoid) and leaky ReLU (Xu et al. (2015)) are variants of the aforementioned functions.

In Algorithm 2, it is evident that each node of the output layer presents a GLM problem (and hence, SLQC) w.r.t. the corresponding weights from $W_2$. We show in Appendices A.2 and A.3 how the entire layer is SLQC w.r.t. $W_2$, by generalizing the definition of SLQC to matrices. In case of $W_1$, while the problem may not directly represent a GLM, we show in Appendix A.3 that our generalized definition of SLQC to functions on matrices allows us to prove that Step 4 of Algorithm 2 is also SLQC w.r.t. $W_1$.

Thus, given a single-layer autoencoder with either sigmoid or ReLU activation functions, DANTE provides an effective alternating minimization strategy that uses SNGD to solve SLQC problems in each alternating step, each of which converges to its respective $\epsilon$-suboptimal solution with high probability, as shown above in Theorem 3.6. Importantly, note that the convergence rate of SNGD depends on the $\kappa$ parameter. Whereas the GLM error function with sigmoid activation has $\kappa = e^W$ Hazan et al. (2015), we obtain $\kappa = \frac{2b^3W}{a}$ (i.e. linear in $W$) for the generalized ReLU setting, which is an exponential improvement. This is significant as in Theorem 3.6, the number of iterations $T$ depends on $\kappa^2$. This shows that SNGD offers accelerated convergence with generalized ReLU GLMs (introduced in this work) when compared to sigmoid GLMs.

### 3.4 EXTENDING TO A MULTI-LAYER AUTOENCODER

In the previous sections, we illustrated how a single hidden-layer autoencoder can be cast as a set of SLQC problems and proposed an alternating minimization method, DANTE. This approach can be generalized to deep autoencoders by considering the greedy layer-wise approach to training a neural network (Bengio et al. (2007)). In this approach, each pair of layers of a deep stacked autoencoder is successively trained in order to obtain the final representation. Each pair of layers considered in this paradigm is a single hidden-layer autoencoder, which can be cast as pairs of SLQC problems that can be trained using DANTE. Therefore, training a deep autoencoder using greedy layer-wise approach can be modeled as a series of SLQC problem pairs. Algorithm 3 summarizes the proposed approach to use DANTE for a deep autoencoder, and Figure 1 illustrates the approach.

Note that it may be possible to use other schemes to use DANTE for multi-layer autoencoders such as a round-robin scheme, where each layer is trained separately one after the other in the sequence in which the layers appear in the network.

## 4 EXPERIMENTS AND RESULTS

We validated DANTE by training autoencoders on an expanded $32 \times 32$ variant of the standard MNIST dataset (LeCun et al. (1998)) as well as other datasets from the UCI repository. We also conducted experiments with multi-layer autoencoders, as well as studied with varying number of hidden neurons

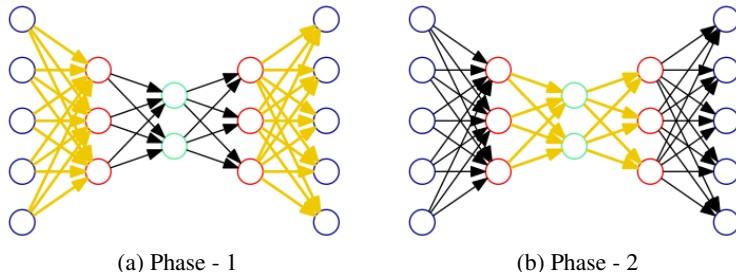

(a) Phase - 1              (b) Phase - 2

Figure 1: An illustration of the proposed multi-layer DANTE (best viewed in color). In each training phase, the outer pairs of weights (shaded in gold) are treated as a single-layer autoencoder to be trained using single-layer DANTE, followed by the inner single-layer auroencoder (shaded in black). These two phases are followed by a finetuning process that may be empirically determined, similar to standard deep autoencoder training.

---

**Algorithm 3:** DANTE for a multi-layer autoencoder

**Input** : Encoder $e$ with weights $\mathbf{U}$, Decoder $d$ with weights $\mathbf{V}$, Number of hidden layers $2n - 1$, Learning rate $\eta$, Stopping threshold $\epsilon$, Number of iterations of alternating minimization $T_{AM}$, initial values $\mathbf{U}^0, \mathbf{V}^0$, minibatch size $b$

1   $t := 1$

2   **for** $l = 1$ *to* $n$ **do**

3      **while** $|f(\boldsymbol{U}^t, \boldsymbol{V}^t) - f(\boldsymbol{U}^{t-1}, \boldsymbol{V}^{t-1})| \geq \epsilon$ *or* $t < T_{AM}$ **do**

4          //Use SNGD for minimizations

         $\mathbf{u}_l^t \leftarrow \arg\min_{\mathbf{u}} \mathbb{E}_{\mathbf{x} \sim \mathcal{D}} \left( d(e(\mathbf{x}, \mathbf{U}_{[1:l-1]}^t \cdot \mathbf{u} \cdot \mathbf{U}_{[l+1:n-1]}^{t-1}), \ \mathbf{V}_{[1:l-1]}^t \cdot \mathbf{V}_{[l:n-1]}^{t-1}) - \mathbf{x} \right)^2$

5          $\mathbf{v}_{n-l}^t \leftarrow \arg\min_{\mathbf{v}} \mathbb{E}_{\mathbf{x} \sim \mathcal{D}} \left( d(e(\mathbf{x}, \mathbf{U}_{[1:l]}^t \cdot \mathbf{U}_{[l+1:n-1]}^{t-1}), \ \mathbf{V}_{[1:n-l-1]}^t \cdot \mathbf{v} \cdot \mathbf{V}_{[n-l+1:n-1]}^{t-1}) - \mathbf{x} \right)^2$

         $t := t + 1$

6      **end**

7   **end**

**Output** : $\mathbf{U}, \mathbf{V}$

---

on single-layer autoencoders. Our experiments on MNIST used the standard benchmarking setup of the dataset[1], with $60,000$ data samples used for training and $10,000$ samples for testing. Experiments were conducted using Torch 7 ( Collobert et al. (2011)).

**Autoencoder with Sigmoid Activation:** A single-layer autoencoder (equivalent to a neural network with one hidden layer) with a sigmoid activation was trained using DANTE as well as standard backprop-SGD (represented as SGD in the results, for convenience) using the standard Mean-Squared Error loss function. The experiments considered 600 hidden units, a learning rate of 0.001, and a minibatch size of 500 (same setup was maintained for SGD and the SNGD used inside DANTE for fair comparison; one could optimize both SGD and SNGD to improve the absolute result values.) We studied the performance by varying the number of hidden neurons, and show those results later in this section. The results are shown in Figure 2a. The figure shows that while DANTE takes slightly (negligibly) longer to reach a local minimum, it obtains a better solution than SGD. (We note that the time taken for the iterations were comparable across both DANTE and backprop-SGD.)

**Autoencoder with ReLU Activation:** Similar to the above experiment, a single-layer autoencoder with a leaky ReLU activation was trained using DANTE and backprop-SGD using the Mean-Squared Error loss function. Once again, the experiments considered 600 units in the hidden layer of the

---

[1]http://yann.lecun.com/exdb/mnist/

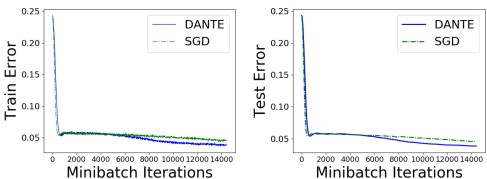 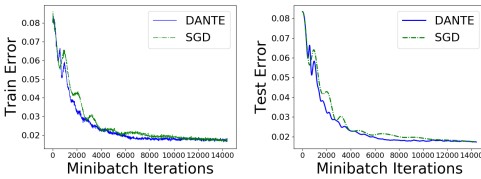

(a) Single-layer autoencoder with Sigmoid activation

(b) Single-layer autoencoder with Generalized ReLU activation

Figure 2: Plots of training and test errors vs training iterations for single hidden-layer autoencoder with Sigmoid (left) and Generalized ReLU (right) activations for both DANTE and SGD.

|  | DANTE | SGD |
|---|---|---|
| MNIST | 93.6% | 92.44% |
| ionosphere | 92.45% | 96.22% |
| svmguide4 | 87.65% | 70.37% |
| USPS | 90.43% | 89.49% |
| vehicle | 77.02% | 74.80% |

Figure 3: Reconstructions using autoencoder models with ReLU activation. *Top:* Original Images; *Middle:* Model trained using DANTE; *Bottom:* Model trained using Backprop-SGD.

Table 1: Classification accuracies using ReLU autoencoder features on different datasets

autoencoder, a leakiness parameter of 0.01 for the leaky ReLU, a learning rate of 0.001, and a minibatch size of 500. The results are shown in Figure 2b. The results for ReLU showed an improvement, and DANTE was marginally better than back-prop SGD across the iterations (as shown in the figure).

In Figure 3, we also show the reconstructions obtained by both trained models (DANTE and Backprop-SGD) for the autoencoder with the Generalized ReLU activation. The model trained using DANTE shows comparable performance as a model trained by SGD under the same settings, in this case. We also conducted experiments to study the effectiveness of the feature representations learned using the models trained using DANTE and SGD in the same setting. After training, we passed the dataset through the autoencoder, extracted the hidden layer representations, and then trained a linear SVM. The classification accuracy results using the hidden representations are given in Table 1. The table clearly shows the competitive performance of DANTE on this task.

**Experiments on other datasets:** We also studied the performance of DANTE on other standard datasets [2], viz. Ionosphere (34 dimensions, 351 datapoints), SVMGuide4 (10 dimensions, 300 datapoints), Vehicle (18 dimensions, 846 datapoints), and USPS (256 dimensions, 7291 datapoints).

[2] https://www.csie.ntu.edu.tw/~cjlin/libsvmtools/datasets/

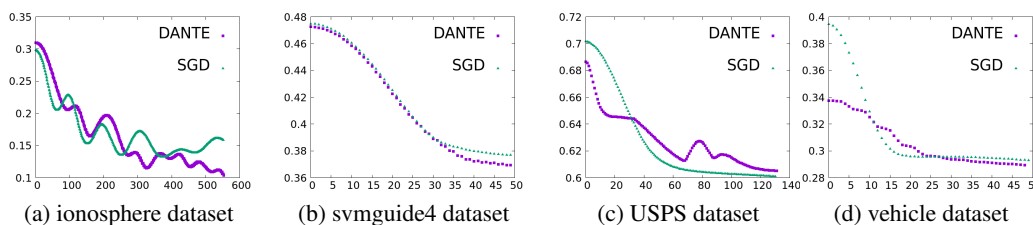

(a) ionosphere dataset    (b) svmguide4 dataset    (c) USPS dataset    (d) vehicle dataset

Figure 4: Comparison of DANTE vs Backprop-SGD on other datasets from the UCI repository. The $x$-axis on all figures is the number of mini-batch iterations and $y$-axis denotes test error, which shows the generalization performance. (Best viewed in color; DANTE = purple, SGD = green)

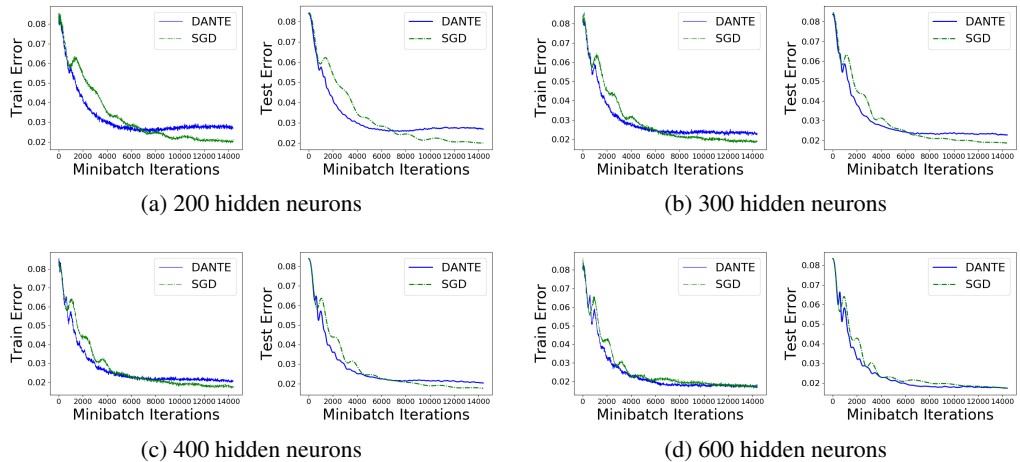

(a) 200 hidden neurons        (b) 300 hidden neurons

(c) 400 hidden neurons        (d) 600 hidden neurons

Figure 5: Plots of training and test error vs training iterations on a single-layer autoencoder with generalized ReLU activation, with varying number of nodes in the hidden layer.

Figure 4 and Table 1 show the performance of the proposed method vs SGD on the abovementioned datasets. It can be seen that DANTE once again demonstrates competitive performance across the datasets, presenting its capability as a viable alternative for standard backprop-SGD.

**Varying Number of Hidden Neurons:** Given the decomposable nature of the proposed solution to learning autoencoders, we also studied the effect of varying hyperparameters across the layers, in particular, the number of hidden neurons in a single-layer autoencoder. The results of these experiments are shown in Figure 5. The plots show that when the number of hidden neurons is low, DANTE reaches its minumum value much sooner (considering this is a subgradient method, one can always choose the best iterate over training) than SGD, although SGD finds a slightly better solution. However, when the number of hidden neurons increases, DANTE starts getting consistently better. This can be attributed to the fact that the subproblem is relatively more challenging for an alternating optimization setting when the number of hidden neurons is lesser.

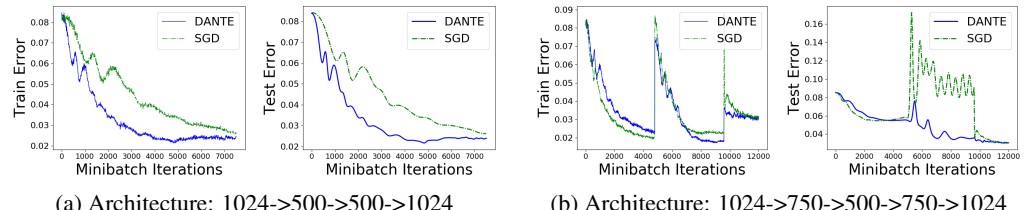

(a) Architecture: 1024->500->500->1024      (b) Architecture: 1024->750->500->750->1024

Figure 6: Plots of training error and test error vs training iterations for multi-layer autoencoders with generalized (leaky) ReLU activations for both DANTE and SGD.

**Multi-Layer Autoencoder:** We also studied the performance of the proposed multi-layer DANTE method (Algorithm 3) for the MNIST dataset. Figure 6 shows the results obtained by stacking two single-layer autoencoders, each with the generalized (leaky) ReLU activation (note that a two single-layer autoencoder corresponds to 4 layers in the overall network, as mentioned in the architecture on the figure). The figure shows promising performance for DANTE in this experiment. Note that Figure 6b shows two spikes: one when the training for the next pair of layers in the autoencoder begins, and another when the end-to-end finetuning process is done. This is not present in Figure 6a, since the $500 \rightarrow 500$ layer in between is only randomly initialized, and is not trained using DANTE or SGD.

## 5 CONCLUSIONS AND FUTURE WORK

In this work, we presented a novel methodology, Deep AlterNations for Training autoEncoders (DANTE), to efficiently train autoencoders using alternating minimization, thus providing an effective alternative to backpropagation. We formulated the task of training each layer of an autoencoder as a Strictly Locally Quasi-Convex (SLQC) problem, and leveraged recent results to use Stochastic Normalized Gradient Descent (SNGD) as an effective method to train each layer of the autoencoder. While recent work was restricted to using sigmoidal activation functions, we introduced a new generalized ReLU activation function, and showed that a GLM with this activation function also satisfies the SLQC property, thus allowing us to expand the applicability of the proposed method to autoencoders with both sigmoid and ReLU family of activation functions. In particular, we extended the definitions of local quasi-convexity to use subgradients in order to prove that the GLM with generalized ReLU activation is $\left(\epsilon, \frac{b^3 W}{2a}, \mathbf{w}^*\right) - SLQC$, which improves the convergence bound for SLQC in the GLM with the generalized ReLU (as compared to a GLM with sigmoid). We also showed how DANTE can be extended to train multi-layer autoencoders. We empirically validated DANTE with both sigmoidal and ReLU activations on standard datasets as well as in a multi-layer setting, and observed that it provides a competitive alternative to standard backprop-SGD, as evidenced in the experimental results.

**Future Work and Extensions.**   DANTE can not only be used to train autoencoders, but can be extended to train standard multi-layer neural networks too. One could use DANTE to train a neural network layer-wise in a round robin fashion, and then finetune end-to-end using backprop-SGD. In case of autoencoders with tied weights, one could use DANTE to learn the weights of the required layers, and then finetune end-to-end using a method such as SGD. Our future work will involve a more careful study of the proposed method for deeper autoencoders, including the settings mentioned above, as well as in studying performance bounds for the end-to-end alternating minimization strategy for the proposed method.

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

# A    APPENDICES

## A.1    NOISY GLM WITH GENERALIZED RELU ACTIVATION FUNCTION

The theorem below is a continuation of the discussion in Section 3.3 (see Theorem 3.5). We prove this result below.

**Theorem A.1.** *In the noisy GLM with generalized ReLU activation, assuming $||\mathbf{w}^*|| \leq W$, given $w \in B(0, W)$, then with probability $\geq 1 - \delta$ after $m \geq \frac{288b^4 W^2}{a^2(1-W^{-1})^2 \epsilon^2} log(1/\delta)/\epsilon^2$ samples, $\hat{err}(\mathbf{w})$ is $\left( \epsilon, \frac{2b^3 W}{a}, \mathbf{w}^* \right) - SLQC$ in $\mathbf{w}$.*

*Proof.* Here, $\forall i, y_i \in [0, 1]$, the following holds:

$$y_i = \phi \langle \mathbf{w}^*, \mathbf{x} \rangle + \xi_i \tag{6}$$

where $\{\xi_i\}_{i=1}^m$ are zero mean, independent and bounded random variables, i.e. $\forall i \in [m], ||\xi_i|| \leq 1$. Then, $\hat{err}_m(\mathbf{w})$ may be written as follows (expanding $y_i$ as in Eqn 6):

$$
\begin{aligned}
\hat{err}_m(\mathbf{w}) &= \frac{1}{m} \sum_{i=1}^m (y_i - \phi \langle \mathbf{w}, \mathbf{x}_i \rangle))^2 \\
&= \frac{1}{m} \Big( \sum_{i=1}^m (\phi \langle \mathbf{w}^*, \mathbf{x}_i \rangle - \phi \langle \mathbf{w}, \mathbf{x}_i \rangle)^2 \\
&\quad + \sum_{i=1}^m 2\xi_i(\phi \langle \mathbf{w}^*, \mathbf{x}_i \rangle - \phi \langle \mathbf{w}, \mathbf{x}_i \rangle) + \sum_{i=1}^m \xi_i^2 \Big)
\end{aligned}
$$

Therefore, we also have (by definition of noisy GLM in Defn 3.2):

$$
\begin{aligned}
\hat{err}_m(\mathbf{w}) - \hat{err}_m(\mathbf{w}^*) &= \frac{1}{m} \sum_{i=1}^m (\phi \langle \mathbf{w}^*, \mathbf{x}_i \rangle - \phi \langle \mathbf{w}, \mathbf{x}_i \rangle)^2 \\
&\quad + \frac{1}{m} \sum_{i=1}^m 2\xi_i(\phi \langle \mathbf{w}^*, \mathbf{x}_i \rangle - \phi \langle \mathbf{w}, \mathbf{x}_i \rangle)
\end{aligned}
$$

Consider $||\mathbf{w}|| \leq W$ such that $\hat{err}_m(\mathbf{w}) - \hat{err}_m(\mathbf{w}^*) \geq \epsilon$. Also, let $\mathbf{v}$ be a point $\epsilon/\kappa$-close to minima $\mathbf{w}^*$ with $\kappa = \frac{2b^3 W}{a}$. Let $g$ be the subgradient of the generalized ReLU activation and $G$ be the subgradient of $\hat{err}_m(\mathbf{w})$, as before. Then:

$$
\begin{aligned}
&\langle G(\mathbf{w}), \mathbf{w} - \mathbf{v} \rangle \\
&= \frac{2}{m} \sum_{i=1}^m g \langle \mathbf{w}, \mathbf{x}_i \rangle \left( \phi \langle \mathbf{w}, \mathbf{x}_i \rangle - y_i \right) \langle \mathbf{x}_i, (\mathbf{w} - \mathbf{v}) \rangle \\
&= \frac{2}{m} \sum_{i=1}^m g \langle \mathbf{w}, \mathbf{x}_i \rangle \left( \phi \langle \mathbf{w}, \mathbf{x}_i \rangle - \phi \langle \mathbf{w}^*, \mathbf{x}_i \rangle - \xi_i \right) && \text{(Step 1)} \\
&\quad\quad\quad [\langle \mathbf{x}_i, \mathbf{w} - \mathbf{w}^* \rangle + \langle \mathbf{x}_i, \mathbf{w}^* - \mathbf{v} \rangle] \\
&\geq \frac{2b^{-1}}{m} \sum_{i=1}^m g \langle \mathbf{w}, \mathbf{x}_i \rangle (\phi \langle \mathbf{w}^*, \mathbf{x}_i \rangle - \phi \langle \mathbf{w}, \mathbf{x}_i \rangle)^2 \\
&\quad - \frac{2}{m} \sum_{i=1}^m g \langle \mathbf{w}, \mathbf{x}_i \rangle \xi_i (\langle \mathbf{w}, \mathbf{x}_i \rangle - \langle \mathbf{w}^*, \mathbf{x}_i \rangle) && \text{(Step 2)} \\
&\quad + \frac{2}{m} \sum_{i=1}^m g \langle \mathbf{w}, \mathbf{x}_i \rangle (\phi \langle \mathbf{w}, \mathbf{x}_i \rangle - \phi \langle \mathbf{w}^*, \mathbf{x}_i \rangle - \xi_i) \langle \mathbf{w}^* - \mathbf{v}, \mathbf{x}_i \rangle)
\end{aligned}
$$

$$\geq \frac{2b^{-1}}{m} \sum_{i=1}^{m} g\langle \mathbf{w}, \mathbf{x}_i \rangle (\phi\langle \mathbf{w}^*, \mathbf{x}_i \rangle - \phi\langle \mathbf{w}, \mathbf{x}_i \rangle)^2$$

$$- \frac{2}{m} \sum_{i=1}^{m} g\langle \mathbf{w}, \mathbf{x}_i \rangle \xi_i (\langle \mathbf{w}, \mathbf{x}_i \rangle - \langle \mathbf{w}^*, \mathbf{x}_i \rangle) - 2\frac{\epsilon b^2}{\kappa} (||\mathbf{w} - \mathbf{w*}|| + \frac{1}{m} \sum_{i=1}^{m} |\xi_i|) \tag{Step 3}$$

$$= \frac{2b^{-1}}{m} \sum_{i=1}^{m} a[(\phi\langle \mathbf{w}^*, \mathbf{x}_i \rangle - \phi\langle \mathbf{w}, \mathbf{x}_i \rangle)^2$$

$$- 2\xi_i (\phi\langle \mathbf{w}, \mathbf{x}_i \rangle - \phi\langle \mathbf{w}^*, \mathbf{x}_i \rangle)]$$

$$- \frac{2}{m} \sum_{i=1}^{m} [g\langle \mathbf{w}, \mathbf{x}_i \rangle (\xi_i (\langle \mathbf{w}, \mathbf{x}_i \rangle - \langle \mathbf{w}^*, \mathbf{x}_i \rangle)) \tag{Step 4}$$

$$- 2ab^{-1}\xi_i (\phi\langle \mathbf{w}, \mathbf{x}_i \rangle - \phi\langle \mathbf{w}^*, \mathbf{x}_i \rangle)] - 2\frac{\epsilon b^2}{\kappa} (||\mathbf{w} - \mathbf{w*}|| + \frac{1}{m} \sum_{i=1}^{m} |\xi_i|)$$

$$\geq 2ab^{-1}\epsilon - 2\frac{\epsilon b^2}{\kappa} (||\mathbf{w} - \mathbf{w*}|| + \frac{1}{m} \sum_{i=1}^{m} |\xi_i|) + \frac{1}{m} \sum_{i=1}^{m} \xi_i \lambda_i(\mathbf{w}) \tag{Step 5}$$

$$\geq 2ab^{-1}\epsilon - ab^{-1}W^{-1}\epsilon (||\mathbf{w} - \mathbf{w*}|| + \frac{1}{m} \sum_{i=1}^{m} |\xi_i|) + \frac{1}{m} \sum_{i=1}^{m} \xi_i \lambda_i(\mathbf{w}) \tag{Step 6}$$

$$\geq 2ab^{-1}\epsilon - ab^{-1}\epsilon(1 + W^{-1}) + \frac{1}{m} \sum_{i=1}^{m} \xi_i \lambda_i(\mathbf{w}) \tag{Step 7}$$

$$\geq -ab^{-1}\epsilon W^{-1} + \frac{1}{m} \sum_{i=1}^{m} \xi_i \lambda_i(\mathbf{w}) \tag{Step 8}$$

Here, $\lambda_i(\mathbf{w}) = 2g\langle \mathbf{w}, \mathbf{x}_i \rangle (\langle \mathbf{w}, \mathbf{x}_i \rangle - \langle \mathbf{w}^*, \mathbf{x}_i \rangle) - 4ab^{-1}(\phi\langle \mathbf{w}, \mathbf{x}_i \rangle - \phi\langle \mathbf{w}^*, \mathbf{x}_i \rangle)$, and
$|\xi_i \lambda_i(\mathbf{w})| \leq 2b(|\langle \mathbf{w}, \mathbf{x}_i \rangle - \langle \mathbf{w}^*, \mathbf{x}_i \rangle| + 4ab^{-1}|\phi\langle \mathbf{w}, \mathbf{x}_i \rangle - \phi\langle \mathbf{w}^*, \mathbf{x}_i \rangle|) \leq 2b(3|\langle \mathbf{w}, \mathbf{x}_i \rangle - \langle \mathbf{w}^*, \mathbf{x}_i \rangle|) \leq 2b(6W) = 12bW$

The above proof uses arguments similar to the proof for the idealized GLM (please see the lines after the proof of Theorem 3.4, viz. the $b$-Lipschitzness of the generalized ReLU, and the problem setup). Now, when

$$\frac{1}{m} \sum_{i=1}^{m} \xi \lambda_i(\mathbf{w}) \geq ab^{-1}W^{-1}\epsilon$$

our model is SLQC. By simply using the Hoeffding's bound, we get that the theorem statement holds for $m \geq \frac{288b^4 W^4}{a^2 \epsilon^2} log(1/\delta)/\epsilon^2$. □

## A.2 Viewing the Outer Layer of an Autoencoder as a Set of GLMs

Given an (unknown) distribution $\mathcal{D}$, let the layer be characterized by a linear operator $W \in \mathbb{R}^{d \times d'}$ and a non-linear activation function defined by $\phi : \mathbb{R} \to \mathbb{R}$. Let the layer output be defined by $\phi\langle W, \mathbf{x} \rangle$, where $\mathbf{x} \in \mathbb{R}^d$ is the input, and $\phi$ is used element-wise in this function.

Consider the mean squared error loss, commonly used in autoencoders, given by:

$$\min_{W} \quad err(W) = \min_{W} \quad \mathbb{E}_{\mathbf{x} \sim \mathcal{D}} ||\phi\langle W, \mathbf{x} \rangle - \mathbf{y}||_2^2$$

$$= \min_{W} \quad \mathbb{E}_{\mathbf{x} \sim \mathcal{D}} || \sum_{i=1}^{d'} \phi\langle W_{:,i}, \mathbf{x} \rangle - \mathbf{y}_i ||_2^2$$

$$= \min_{W} \quad \sum_{i=1}^{d'} \mathbb{E}_{\mathbf{x} \sim \mathcal{D}} ||\phi\langle W_{:,i}, \mathbf{x} \rangle - \mathbf{y}_i||_2^2$$

$$= \sum_{i=1}^{d'} \min_{W} \quad \mathbb{E}_{\mathbf{x} \sim \mathcal{D}} ||\phi\langle W_{:,i}, \mathbf{x} \rangle - \mathbf{y}_i||_2^2$$

Each of these sub-problems above is a GLM, which can be solved effectively using SNGD as seen in Theorem 3.6, which we leverage in this work.

### A.3 LOCAL QUASI-CONVEXITY OF THE AUTOENCODER

In Algorithm 2, while it is evident that each of the problems in Step 3 is a GLM and hence, SLQC, w.r.t. the corresponding parameters in $W_2$, we show here that the complete layer in Step 3 is also SLQC w.r.t. $W_2$, as well as show that the problem in Step 4 is SLQC w.r.t. $W_1$. We begin with the definition of SLQC for matrices, which is defined using the Frobenius inner product.

**Definition A.2** (*Local-Quasi-Convexity for Matrices*). Let $\mathbf{x}, \mathbf{z} \in \mathbb{R}^{d \times d'}, \kappa, \epsilon > 0$ and let $f : \mathbb{R}^{d \times d'} \to \mathbb{R}$ be a differentiable function. Then $f$ is said to be $(\epsilon, \kappa, \mathbf{z})$-Strictly-Locally-Quasi-Convex (SLQC) in $\mathbf{x}$, if at least one of the following applies:

1. $f(\mathbf{x}) - f(\mathbf{z}) \leq \epsilon$

2. $\|\nabla f(\mathbf{x})\| > 0$, and $\forall \mathbf{y} \in \mathbb{B}(\mathbf{z}, \epsilon/\kappa), \ Tr(\nabla f(\mathbf{x})^T(\mathbf{y} - \mathbf{x})) \leq 0$

where $\mathbb{B}(\mathbf{z}, \epsilon/\kappa)$ refers to a ball centered at $\mathbf{z}$ with radius $\epsilon/\kappa$.

We now prove that the $\hat{err}(\mathbf{W})$ of a multi-output single-layer neural network is indeed SLQC in $\mathbf{W}$. This corresponds to proving that the one-hidden layer autoencoder problem is SLQC in $W_2$. We then go on to prove that a two layer single-output neural network is SLQC in the first layer $W_1$, which can be trivially extended using the basic idea seen in Theorem A.4 to show that the one hidden-layer autoencoder problem is also SLQC in $W_1$.

**Theorem A.3.** *Let an idealized single-layer multi-output neural network be characterized by a linear operator $\mathbf{W} \in \mathbb{R}^{d \times d'} = [\mathbf{w}_1 \ \mathbf{w}_2 \ \cdots \ \mathbf{w}_{d'}]$ and a generalized ReLU activation function $\phi : \mathbb{R} \to \mathbb{R}$. Let the output of the layer be $\phi\langle\mathbf{W}, \mathbf{x}\rangle$ where $\mathbf{x} \in \mathbb{R}^d$ is the input, and $\phi$ is applied element-wise. Assuming $||\mathbf{W}^*|| \leq C$, $\hat{err}(\mathbf{W})$ is $\left(\epsilon, \frac{2b^3 C}{a}, \mathbf{W}^*\right) - SLQC$ in $\mathbf{W}$ for all $\mathbf{W} \in \mathbb{B}_d(0, C)$.*

*Proof.* Consider:

$$\langle G(\mathbf{W}), \mathbf{W} - \mathbf{V}\rangle_F$$

$$= \sum_{j=1}^{d'} \langle G(\mathbf{w_j}), \mathbf{w_j} - \mathbf{v_j}\rangle \qquad \text{(By defn of Frobenius inner product)}$$

$$= \frac{2}{m} \sum_{i=1}^{m} \sum_{j=1}^{d'} (\phi\langle\mathbf{w_j}, \mathbf{x_i}\rangle - y_{ij}) \langle \frac{\partial(\phi\langle\mathbf{w_j}, \mathbf{x_i}\rangle)}{\partial \mathbf{w_j}}, (\mathbf{w_j} - \mathbf{v_j})\rangle_F \qquad \text{(Step 1)}$$

$$= \frac{2}{m} \sum_{i=1}^{m} \sum_{j=1}^{d'} g(\mathbf{w_j}, \mathbf{x}_i)(\phi\langle\mathbf{w_j}, \mathbf{x_i}\rangle - y_{ij}) [\langle\mathbf{x}_i, \mathbf{w_j} - \mathbf{w_j^*}\rangle + \langle\mathbf{x}_i, \mathbf{w_j^*} - \mathbf{v_j}\rangle_F] \qquad \text{(Step 2)}$$

$$\geq \frac{2}{m} \sum_{i=1}^{m} \sum_{j=1}^{d'} g\langle\mathbf{w_j}, \mathbf{x}_i\rangle \Big[b^{-1}\left(\phi\langle\mathbf{w_j}, \mathbf{x_i}\rangle - \phi\langle\mathbf{w_j^*}, \mathbf{x_i}\rangle\right)^2 + \left(\phi\langle\mathbf{w_j}, \mathbf{x_i}\rangle - \phi\langle\mathbf{w_j^*}\mathbf{x_i}\rangle\right)\langle\mathbf{x}_i, \mathbf{w_j^*} - \mathbf{v_j}\rangle\Big]$$

$$\text{(Step 3)}$$

$$\geq \frac{2}{m} \sum_{i=1}^{m} \sum_{j=1}^{d'} g\langle\mathbf{w_j}, \mathbf{x}_i\rangle \Big[(b^{-1}\left(\phi\langle\mathbf{w_j}, \mathbf{x_i}\rangle - \phi\langle\mathbf{w_j^*}, \mathbf{x_i}\rangle\right)^2 - |\phi\langle\mathbf{w_j}, \mathbf{x_i}\rangle - \phi\langle\mathbf{w_j^*}, \mathbf{x_i}\rangle|||\mathbf{x}_i||||\mathbf{w_j^*} - \mathbf{v_j}||\Big]$$

$$\geq \frac{2}{m} \sum_{i=1}^{m} \sum_{j=1}^{d'} \Big[ab^{-1}\left(\phi\langle\mathbf{w_j}, \mathbf{x_i}\rangle - \phi\langle\mathbf{w_j^*}, \mathbf{x_i}\rangle\right)^2 - b|\phi\langle\mathbf{w_j}, \mathbf{x_i}\rangle - \phi\langle\mathbf{w_j^*}, \mathbf{x_i}\rangle|||\mathbf{x}_i||||\mathbf{w_j^*} - \mathbf{v_j}||\Big]$$

$$\text{(Step 4)}$$

$$\geq \frac{2}{m} \sum_{i=1}^{m} \sum_{j=1}^{d'} \Big[ab^{-1}\left(\phi\langle\mathbf{w_j}, \mathbf{x_i}\rangle - \phi\langle\mathbf{w_j^*}, \mathbf{x_i}\rangle\right)^2 - b^2||\langle\mathbf{w_j}, \mathbf{x_i}\rangle - \langle\mathbf{w_j^*}, \mathbf{x_i}\rangle||\frac{\epsilon}{\kappa}||\mathbf{x}_i||\Big] \qquad \text{(Step 5)}$$

$$\geq 2ab^{-1}d'\epsilon - \frac{ad'\epsilon}{bC}||\langle \mathbf{w}, \mathbf{x}_i\rangle - \langle \mathbf{w}^*, \mathbf{x}_i\rangle|||\mathbf{x}_i|| \tag{Step 6}$$

$$\geq ab^{-1}d'\epsilon(2 - \frac{1}{C}||\mathbf{w} - \mathbf{w}^*||||\mathbf{x}_i||^2)$$

$$\geq 0 \qquad\qquad\qquad\qquad\qquad\qquad\qquad\qquad \square$$

The remainder of the proof proceeds precisely as in Theorem 3.4.

**Theorem A.4.** *Let an idealized two-layer neural network be characterized by a linear operator* $\mathbf{w_1} \in \mathbb{R}^{d \times d'}$, $\mathbf{w_2} \in \mathbb{R}^{d'}$ *and generalized ReLU activation functions* $\phi_1 : \mathbb{R}^{d'} \to \mathbb{R}^{d'}$, $\phi_2 : \mathbb{R} \to \mathbb{R}$ *with a setting similar to Equation 5. Assuming* $||\mathbf{w_1^*}|| \leq W_1$, $||\mathbf{w_2^*}|| \leq W_2$, $\hat{err}(\mathbf{w_1}, \mathbf{w_2})$ *is* $\left(\epsilon, \frac{2b^5 W_2^2 W_1}{a}, \mathbf{w_1^*}\right) - SLQC$ *in* $\mathbf{w_1}$ *for all* $\mathbf{w_1} \in \mathbb{B}_d(0, W_1)$.

*Proof.* Let $\|f(\mathbf{w_1}; \mathbf{w_2}; \mathbf{x}) - \mathbf{x}\|_2^2 = \|\phi_2\langle \mathbf{w_2}, \phi_1\langle \mathbf{w_1}, \mathbf{x}\rangle\rangle - \mathbf{x}\|_2^2$ and $\langle \cdot \rangle_F$ be the Frobenius inner product.

$$\langle G(\mathbf{w_1}), \mathbf{w_1} - \mathbf{v_1}\rangle_F$$
$$= \frac{2}{m}\sum_{i=1}^m (\phi_2\langle \mathbf{w_2}, \phi_1\langle \mathbf{w_1}, \mathbf{x_i}\rangle\rangle - y_i) \langle \frac{\partial(\phi_2\langle \mathbf{w_2}, \phi_1\langle \mathbf{w_1}, \mathbf{x_i}\rangle\rangle)}{\partial \mathbf{w}_1}, (\mathbf{w_1} - \mathbf{v_1})\rangle_F \tag{Step 1}$$

Using chain rule, we can simplify $\frac{\partial(\phi_2\langle \mathbf{w_2}, \phi_1\langle \mathbf{w_1}, \mathbf{x}\rangle\rangle)}{\partial \mathbf{w}_1}$ as

$$\left[\frac{\partial(\phi_2\langle \mathbf{w_2}, \phi_1\langle \mathbf{w_1}, \mathbf{x}\rangle\rangle)}{\partial \mathbf{w}_1}\right]^T = \frac{\partial(\phi_2\langle \mathbf{w_2}, \phi_1\langle \mathbf{w_1}, \mathbf{x}\rangle\rangle)}{\partial \langle \mathbf{w_2}, \phi_1\langle \mathbf{w_1}, \mathbf{x}\rangle\rangle} \cdot \left[\frac{\partial\langle \mathbf{w_2}, \phi_1\langle \mathbf{w_1}, \mathbf{x}\rangle\rangle^T}{\partial \phi_1\langle \mathbf{w_1}, \mathbf{x}\rangle} \cdot \frac{\partial\phi_1\langle \mathbf{w_1}, \mathbf{x}\rangle^T}{\partial\langle \mathbf{w_1}, \mathbf{x_i}\rangle}\right]^T \cdot \left[\frac{\partial\langle \mathbf{w_1}, \mathbf{x}\rangle}{\partial \mathbf{w}_1}\right]^T$$
$$= g_2(\mathbf{w_1}, \mathbf{w_2}, x) \cdot g_1(\mathbf{w_1}, x) \cdot \mathbf{w_2} \cdot \mathbf{x}^T \tag{Let}$$

Continuing from Step 1:

$$= \frac{2}{m}\sum_{i=1}^m g_2(\mathbf{w_1}, \mathbf{w_2}, \mathbf{x_i}) (\phi_2\langle \mathbf{w_2}, \phi_1\langle \mathbf{w_1}, \mathbf{x_i}\rangle\rangle - y_i) \langle \mathbf{x}_i \mathbf{w}_2^T g_1(\mathbf{w_1}, \mathbf{x}_i)^T, (\mathbf{w_1} - \mathbf{v_1})\rangle_F$$

$$= \frac{2}{m}\sum_{i=1}^m g_2(\mathbf{w_1}, \mathbf{w_2}, \mathbf{x_i}) (\phi_2\langle \mathbf{w_2}, \phi_1\langle \mathbf{w_1}, \mathbf{x_i}\rangle\rangle - y_i) [\langle \mathbf{x}_i \mathbf{w}_2^T g_1(\mathbf{w_1}, \mathbf{x}_i)^T, (\mathbf{w_1} - \mathbf{w_1^*})\rangle_F$$
$$+ \langle \mathbf{x}_i \mathbf{w}_2^T g_1(\mathbf{w_1}, \mathbf{x}_i)^T, (\mathbf{w_1^*} - \mathbf{v_1})\rangle_F] \tag{Step 2}$$

$$= \frac{2}{m}\sum_{i=1}^m g_2(\mathbf{w_1}, \mathbf{w_2}, \mathbf{x_i}) (\phi_2\langle \mathbf{w_2}, \phi_1\langle \mathbf{w_1}, \mathbf{x_i}\rangle\rangle - y_i) [Tr(g_1(\mathbf{w_1}, \mathbf{x}_i) \mathbf{w}_2 \mathbf{x}_i^T (\mathbf{w_1} - \mathbf{w_1^*}))$$
$$+ Tr(g_1(\mathbf{w_1}, \mathbf{x}_i) \mathbf{w}_2 \mathbf{x}_i^T (\mathbf{w_1^*} - \mathbf{v_1})]$$

$$= \frac{2}{m}\sum_{i=1}^m g_2(\mathbf{w_1}, \mathbf{w_2}, \mathbf{x_i}) (\phi_2\langle \mathbf{w_2}, \phi_1\langle \mathbf{w_1}, \mathbf{x_i}\rangle\rangle - y_i) [Tr(g_1(\mathbf{w_1}, \mathbf{x}_i) \mathbf{w}_2 \mathbf{x}_i^T \mathbf{w_1})$$
$$- Tr(g_1(\mathbf{w_1^*}, \mathbf{x}_i) \mathbf{w}_2 \mathbf{x}_i^T \mathbf{w_1^*}) + Tr((g_1(\mathbf{w_1^*}, \mathbf{x}_i) - g_1(\mathbf{w_1}, \mathbf{x}_i)) \mathbf{w}_2 \mathbf{x}_i^T \mathbf{w_1^*})$$
$$+ Tr(g_1(\mathbf{w_1}, \mathbf{x}_i) \mathbf{w}_2 \mathbf{x}_i^T (\mathbf{w_1^*} - \mathbf{v_1}))] \tag{Step 3}$$

In order to convert the above terms into a more familiar form, we begin with the following observation:

$$\langle \mathbf{w_2}, \phi_1\langle \mathbf{w}, \mathbf{x}\rangle\rangle_F = Tr(g_1(\mathbf{w}, \mathbf{x}) \mathbf{w}_2 \mathbf{x}^T(\mathbf{w}))$$

Also, note that $g_1(\mathbf{w}, \mathbf{x})$ is a diagonal $d' \times d'$ matrix consisting of $a$'s and $b$'s on the diagonal:

$$(\langle \mathbf{w_2}, \phi_1\langle \mathbf{w_1}, \mathbf{x}\rangle\rangle - \langle \mathbf{w_2}, \phi_1\langle \mathbf{w}, \mathbf{x}\rangle\rangle) = Tr(g_1(\mathbf{w_1}, \mathbf{x}) \mathbf{w}_2 \mathbf{x}^T \mathbf{w_1}) - Tr(g_1(\mathbf{w}, \mathbf{x}) \mathbf{w}_2 \mathbf{x}^T \mathbf{w})$$

Therefore, on setting $\mathbf{w} = \mathbf{w}_1^*$ and using the fact that the generalized ReLU is $b$-Lipschitz and monotonically increasing, we have:

$$
\begin{aligned}
(\phi_2\langle \mathbf{w}_2, \phi_1\langle \mathbf{w}_1, \mathbf{x}\rangle\rangle - \phi_2\langle \mathbf{w}_2, \phi_1\langle\, \mathbf{w_1^*}, \mathbf{x}\rangle\rangle)^2 &\le b(\phi_2\langle \mathbf{w}_2, \phi_1\langle \mathbf{w}_1, \mathbf{x}\rangle\rangle - \phi_2\langle \mathbf{w}_2, \phi_1\langle\, \mathbf{w_1^*}, \mathbf{x}\rangle\rangle) \\
&\qquad (\langle \mathbf{w}_2, \phi_1\langle \mathbf{w}_1, \mathbf{x}\rangle\rangle - \langle \mathbf{w}_2, \phi_1\langle \mathbf{w_1^*}, \mathbf{x}\rangle\rangle) \\
&= b(\phi_2\langle \mathbf{w}_2, \phi_1\langle \mathbf{w}_1, \mathbf{x}\rangle\rangle - \phi_2\langle \mathbf{w}_2, \phi_1\langle\, \mathbf{w_1^*}, \mathbf{x}\rangle\rangle) \\
&\qquad \cdot (Tr(g_1(\mathbf{w_1}, \mathbf{x})\mathbf{w}_2\mathbf{x}^T\mathbf{w_1}) - Tr(g_1(\mathbf{w}, \mathbf{x})\mathbf{w}_2\mathbf{x}^T\mathbf{w_1^*}))
\end{aligned}
$$

Plugging this result into Step 3:

$$
\ge \frac{2}{m}\sum_{i=1}^m g_2(\mathbf{w}_1, \mathbf{w}_2, \mathbf{x}_i)[b^{-1}(\phi_2\langle \mathbf{w}_2, \phi_1\langle \mathbf{w}_1, \mathbf{x_i}\rangle\rangle - \phi_2\langle \mathbf{w}_2, \phi_1\langle\, \mathbf{w_1^*}, \mathbf{x_i}\rangle\rangle))^2
$$
$$
+ (\phi_2\langle \mathbf{w}_2, \phi_1\langle \mathbf{w}_1, \mathbf{x_i}\rangle\rangle - y_i) \cdot Tr((g_1(\mathbf{w}_1^*, \mathbf{x}_i) - g_1(\mathbf{w}_1, \mathbf{x}_i))\mathbf{w}_2\mathbf{x}_i^T\mathbf{w_1^*})
$$
$$
+ (\phi_2\langle \mathbf{w}_2, \phi_1\langle \mathbf{w}_1, \mathbf{x_i}\rangle\rangle - y_i) \cdot Tr(g_1(\mathbf{w}_1, \mathbf{x}_i)\mathbf{w}_2\mathbf{x}_i^T(\mathbf{w_1^*} - \mathbf{v_1}))] \tag{Step 4}
$$

$$
\ge 2ab^{-1}\epsilon + \frac{2}{m}\sum_{i=1}^m g_2(\mathbf{w}_1, \mathbf{w}_2, \mathbf{x}_i)\,(\phi_2\langle \mathbf{w}_2, \phi_1\langle \mathbf{w}_1, \mathbf{x_i}\rangle\rangle - y_i) \tag{Step 5}
$$
$$
\cdot [Tr(g_1(\mathbf{w}_1^*, \mathbf{x}_i)\mathbf{w}_2\mathbf{x}_i^T\mathbf{w}_1^*) - Tr(g_1(\mathbf{w}_1, \mathbf{x}_i)\mathbf{w}_2\mathbf{x}_i^T\mathbf{v_1})]
$$

$$
\ge \frac{2a}{b}\epsilon - \frac{2}{m}\sum_{i=1}^m g_2(\mathbf{w}_1, \mathbf{w}_2, \mathbf{x}_i) \cdot |\,(\phi_2\langle \mathbf{w}_2, \phi_1\langle \mathbf{w}_1, \mathbf{x_i}\rangle\rangle - y_i)\,|\|\mathbf{w}_2\|\|\mathbf{x}_i^T\| \tag{Step 6}
$$
$$
\big[|b|\|\mathbf{w}_1^*\| + b\|\mathbf{v_1}\|\big]
$$

$$
\ge \frac{2a}{b}\epsilon - \frac{2}{m}\sum_{i=1}^m b \cdot |\,(\phi_2\langle \mathbf{w}_2, \phi_1\langle \mathbf{w}_1, \mathbf{x_i}\rangle\rangle - y_i)\,|W_2 \cdot 1 \cdot [b\|\mathbf{w}_1^*\| + b\|\mathbf{v_1}\|] \tag{Step 7}
$$

$$
\ge \frac{2a}{b}\epsilon - \frac{2}{m}\sum_{i=1}^m bW_2 \cdot |\,(\phi_2\langle \mathbf{w}_2, \phi_1\langle \mathbf{w}_1, \mathbf{x_i}\rangle\rangle - y_i)\,| \cdot [b\|\mathbf{w}_1^*\| + b\|\mathbf{v_1}\|] \tag{Step 8}
$$

$$
\ge \frac{2a}{b}\epsilon - \frac{2}{m}\sum_{i=1}^m b^2 W_2 \cdot |\,(\phi_2\langle \mathbf{w}_2, \phi_1\langle \mathbf{w}_1, \mathbf{x_i}\rangle\rangle - y_i)\,| \cdot \|\mathbf{w}_1^* - \mathbf{v_1}\| \tag{Step 9}
$$

$$
= \frac{2a}{b}\epsilon - \frac{2}{m}\sum_{i=1}^m b^2 W_2 \cdot |\phi_2\langle \mathbf{w}_2, \phi_1\langle \mathbf{w}_1, \mathbf{x_i}\rangle\rangle - \phi_2\langle \mathbf{w}_2, \phi_1\langle \mathbf{w}_1^*, \mathbf{x_i}\rangle\rangle| \cdot \|\mathbf{w}_1^* - \mathbf{v_1}\| \tag{Step 10}
$$

$$
\ge \frac{2a}{b}\epsilon - 2b^4 W_2^2 \cdot \|\mathbf{w}_1 - \mathbf{w}_1^*\| \cdot \frac{\epsilon}{\kappa} \tag{Step 11}
$$

$$
= \frac{2a}{b}\epsilon - 2b^4 W_2^2 \frac{\epsilon}{\frac{2b^5 W_2^2 W_1}{a}}\|\mathbf{w}_1 - \mathbf{w}_1^*\| = \frac{a}{b}\epsilon(2 - \frac{1}{W_1}\|\mathbf{w}_1 - \mathbf{w}_1^*\|) \ge 0 \tag{Step 12}
$$

We arrive at Step 6 by using a similar observation as used in Step 4. The remainder of the proof is similar to that of Theorem 3.4.

$\square$

