# OpenReview forum: "Training Autoencoders by Alternating Minimization"
_ICLR.cc/2018/Conference — Reject_

### Official Review · AnonReviewer2 · 2017-11-26
**an attempt of new training method for DNNs**

**Rating:** 6
**Confidence:** 4

**Review:**

After reading the rebuttal:

The authors addressed some of my theoretical questions. I think the paper is borderline, leaning towards accept.

I do want to note my other concerns:

I suspect the theoretical results obtained here are somewhat restricted to the least-squares, autoencoder loss.

And note that the authors show that the proposed algorithm performs comparably to SGD, but not significantly better. The classification result (Table 1) was obtained on the autoencoder features instead of training a classifier on the original inputs. So it is not clear if the proposed algorithm is better for training the classifier, which may be of more interest.

=============================================================

This paper presents an algorithm for training deep neural networks. Instead of computing gradient of all layers and perform updates of all weight parameters at the same time, the authors propose to perform alternating optimization on weights of individual layers.

The theoretical justification is obtained for single-hidden-layer auto-encoders. Motivated by recent work by Hazan et al 2015, the authors developed the local-quasi-convexity of the objective w.r.t. the hidden layer weights for the generalized RELU activation. As a result, the optimization problem over the single hidden layer can be optimized efficiently using the algorithm of Hazan et al 2015. This itself can be a small, nice contribution.

What concerns me is the extension to multiple layers. Some questions are not clear from section 3.4:
1. Do we still have local-quasi-convexity for the weights of each layer, when there are multiple nonlinear layers above it? A negative answer to this question will somewhat undermine the significance of the single-hidden-layer result.

2. Practically, even if the authors can perform efficient optimization of weights in individual layers, when there are many layers, the alternating optimization nature of the algorithm can possibly result in overall slower convergence. Also, since the proposed algorithm still uses gradient based optimizers for each layer, computing the gradient w.r.t. lower layers (closer to the inputs) are still done by backdrop, which has pretty much the same computational cost of the regular backdrop algorithm for updating all layers at the same time. As a result, I am not sure if the proposed algorithm is on par with / faster than the regular SGD algorithm in actual runtime. In the experiments, the authors plotted the training progress w.r.t. the minibatch iterations, I do not know if the minibatch iteration is a proxy for actual runtime (or number of floating point operations).

3. In the experiments, the authors found the network optimized by the proposed algorithm generalize better than regular SGD. Is this result consistent (across dataset, random initializations, etc), and can the authors elaborate the intuition behind?

---

### Official Review · AnonReviewer1 · 2017-11-26
**Some interesting ideas. But, not sure if they are applicable to the autoencoder problem and it is not clear if it outperforms SGD.**

**Rating:** 4
**Confidence:** 4

**Review:**

The authors propose an alternating minimization framework for training autoencoders and encoder-decoder networks. The central idea is that a single encoder-decoder network can be cast as an alternating minimization problem. Each minimization problem is not convex but is quasi-convex and hence one can use stochastic normalized gradient descent to minimize w.r.t. each variable. This leads to the proposed algorithm called DANTE which simply minimizes w.r.t. each variable using stochastic normalized gradient algorithm to minimize w.r.t. each variable The authors start with this idea and introduce a generalized ReLU which is specified via a subgradient function only whose local quasi-convexity properties are established. They then extend these idea to multi-layer encoder-decoder networks by performing greedy layer-wise training and using the proposed algorithms for training each layer. The ideas are interesting, but I have some concerns regarding this work.

Major comments:

1. When dealing with a 2 layer network where there are 2 matrices W_1, W_2 to optimize over. It is not clear to me why optimizing over W_1 is a quasi-convex optimization problem? The authors seem to use the idea that solving a GLM problem is a quasi-convex optimization problem. However, optimizing w.r.t. W_1 is definitely not a GLM problem, since W_1 undergoes two non-linear transformations one via \phi_1 and another via \phi_2. Could the authors justify why minimizing w.r.t. W_1 is still a quasi-convex optimization problem?

2. Theorem 3.4, 3.5 establish  SLQC properties with generalized RELU activations. This is an interesting result, and useful in its own right. However, it is not clear to me why this result is even relevant here. The main application of this paper is autoencoders, which are functions from R^d -> R^d. However, GLMs are functions from R^d ---> R. So, it is not at all clear to me how Theorem 3.4, 3.5 and eventually 3.6 are useful for the autoencoder problem that the authors care about. Yes they are useful if one was doing 2-layer neural networks for binary classification, but it is not clear to me how they are useful for autoencoder problems.

3. Experimental results for classification are not convincing enough. If, one looks at Table 1. SGD outperforms DANTE on ionosphere dataset and is competent with DANTE on MNIST and USPS.

4. The results on reconstruction do not show any benefits for DANTE over SGD (Figure 3). I would recommend the authors to rerun these experiments but truncate the iterations early enough. If DANTE has better reconstruction performance than SGD with fewer iterations then that would be a positive result.

---

### Official Review · AnonReviewer3 · 2017-11-27
**Interesting approach to training Autoencoders**

**Rating:** 7
**Confidence:** 5

**Review:**

In this paper an alternating optimization approach is explored for training Auto Encoders (AEs).
The authors treat each layer as a generalized linear model, and suggest to use the stochastic normalized GD of [Hazan et al., 2015] as the minimization algorithm in each (alternating) phase.
Then they apply the suggested method to several single layer and multi layer AEs, comparing its performance to standard SGD. The paper suggests an interesting approach and provides experimental evidence for its usefulness, especially for multi-layer AEs.


Some comments on the theoretical part:
-The theoretical part is partly misleading. While it is true that every layer can be treated a generalized linear model, the SLQC property only applies for the last layer.
Regarding the intermediate layers, we may indeed treat them as generalized linear models, but with non-monotone activations, and therefore the SLQC property does not apply.
The authors should mention this point.

-Showing that generalized ReLU is SLQC with a polynomial dependence on the domain is interesting.

-It will be interesting if the authors can provide an analysis/relate to some theory related to alternating minimization of bi-quasi-convex objectives. Concretely: Is there any known theory for such objectives? What guarantees can we hope to achieve?


The extension to muti-layer AEs makes sense and seems to works quite well in practice.

The experimental part is satisfactory, and seems to be done in a decent manner.
It will be useful if the authors could relate to the issue of parameter tuning for their algorithm.
Concretely: How sensitive/robust is their approach compared to SGD with respect to hyperparameter misspecification.

---

### Author Response · Authors · 2017-12-23
**Responses to Reviews**

We thank the reviewers for acknowledging our contributions and sharing their feedback. Please find our responses below (we have also uploaded a revised paper draft with the appropriate content based on these responses):

R3: “every layer can be treated a generalized linear model...the SLQC property only applies for the last layer…”
R2: “Do we still have local-quasi-convexity for the weights of each layer, when there are multiple nonlinear layers above it?”

The SLQC property does hold for the intermediate layers as well. It is important to note, however, that the SLQC property holds with respect to the input to the corresponding intermediate layer. (Section 3.3 in the latest version of the paper clarifies this.)

R1: “Experimental results for classification are not convincing enough...Table 1….SGD outperforms….and is competent with DANTE...”
R1: “The results on reconstruction do not show benefits for DANTE (over SGD)…”
R2: “In the experiments, the authors found the network optimized by the proposed algorithm generalize better than regular SGD. Is this result consistent (across dataset, random initializations, etc)?”

We would like to clarify that we propose DANTE (and thus, an alternating minimization strategy) as a competitive alternative to backpropagation-based SGD, and our results corroborate this claim. While this was conveyed in our earlier version too, we have revised any choice of words that may have suggested otherwise. This comparable performance is consistent across all our experiments and studies.

R2: “In the experiments, the authors plotted the training progress w.r.t. the minibatch iterations, I do not know if the minibatch iteration is a proxy for actual runtime (or number of floating point operations).”
Minibatch iterations is in fact a proxy for actual runtime in these experiments, and on measuring the time taken for the experiments in Figure 2, we found the times taken are indeed comparable.

R1: “When dealing with a 2 layer network where there are 2 matrices W_1, W_2 to optimize over. It is not clear to me why optimizing over W_1 is a quasi-convex optimization problem? The authors seem to use the idea that solving a GLM problem is a quasi-convex optimization problem. However, optimizing w.r.t. W_1 is definitely not a GLM problem, since W_1 undergoes two non-linear transformations one via \phi_1 and another via \phi_2. Could the authors justify why minimizing w.r.t. W_1 is still a quasi-convex optimization problem?”

R1: “...autoencoders, which are functions from R^d -> R^d’. However, GLMs are functions from R^d ---> R. So, it is not at all clear to me how Theorem 3.4, 3.5 and eventually 3.6 are useful for the autoencoder problem...”

We have revised Section 3.3 (as well as included additional results in our Appendix) to clarify these questions.

In the DANTE algorithm (Algorithm 2 of paper), it is evident that each node of the output layer presents a GLM problem (and hence, SLQC) w.r.t. the corresponding weights from W_2. We show in Appendices A.2 and A.3 how the entire layer is SLQC w.r.t. W_2, by generalizing the definition of SLQC to matrices. In case of W_1, while the problem may not directly represent a GLM, we show in Appendix A.3 that our generalized definition of SLQC to functions on matrices allows us to prove that Step 4 of Algorithm 2 is also SLQC w.r.t. W_1, thus allowing us to use Theorems 3.4, 3.5 and 3.6 for our formulation.

R3: “It will be interesting if the authors can provide an analysis/relate to some theory related to alternating minimization of bi-quasi-convex objectives. Concretely: Is there any known theory for such objectives? What guarantees can we hope to achieve?”

We are definitely interested in this question ourselves, and this will form an important direction of our future work. To the best of our knowledge, there are no such known guarantees for this setting.

---

> ### Comment · AnonReviewer3 · 2018-01-10
> **Proof of the new theorem A.3 seems to problematic**
>
> Dear Authors,
> I believe that  theorem A.3 that you added regarding  the SLQC property of intermediate layers is incorrect.
>
> I could not spot the mistake in the proof, but I came up with a very simple example showing that SLQC property could not hold,
>
> %%%%%%%%%%%%%%%%%%%%%%%%%%
> Consider the following example:
> -Assume a,b,c,d are scalars.
> -Also assume that we are in the realizable case. Meaning given x its label is set as follows,
>                                  y = max{0,cx}+max{0,dx}
>
> -For a slotion (a,b) and point x we denote,  \phi(a,b,x) = max{0,ax}+max{0,bx}
> -We also denote by Ind{A} the indicator function of an event A.
>
> -Now consider a solution (a,b)
> Then the error of this solution at a point x will be, error = (\phi(a,b, x) - y)^2,
> -The derivative at (a,b) is therefore,
> G = (\phi(a,b, x) - y) * ( Ind{ax>0}x, Ind{bx>0}x)
>
> -Now,
> <G, (a,b) -(c, d) > = (\phi(a,b, x) - y) *(nd{ax>0} *(a-c)*x + Ind{bx>0}*(b-d)*x)
>
> -Lets make the following simplifying assumptions,
> ax>0, bx>0, cx>0, dx<0
> -In this case we get,
>  <G, (a,b) -(c, d) > = ( (a-c)*x + bx)* ( (a-c)*x + (b-d)*x)
>                               = ( (a-c)*x + bx)^2 -dx *  ( (a-c)*x + b*x)
>
> -Now if we assume that (a-c)*x + bx<0 and also dx<  (a-c)*x + bx
> This implies that,   <G, (a,b) -(c, d) > <0
> *And therefore SLQC property does not apply in this case!
> *Note that since dx<0 and bx>0 then then the error = (\phi(a,b, x) - y)^2, might be very large (much larger than \epsilon)
> %%%%%%%%%%%%%%%%%%%%%%%%%%%%%%%%%
>
> Having said that I still believe that the paper is interesting and I keep my score.

---

> > ### Author Response · Authors · 2018-01-13
> > **Response to Reviewer Comment**
> >
> > We thank you for going through our revised draft, and sharing your concern.
> >
> > In our setup for Theorem A.3, we have a single *multi-output* layer, so the label set given x is given by (y being a vector):
> >
> >         y = [ max{0, cx}, max{0,dx} ]
> >
> > Assuming the setup mentioned in the comment with a,b,c and d as scalars, we would then have a setup with one input node, and two output nodes, where (a, b) are the current weights, and (c,d) is the optimal solution. The error is then given by: (\phi(a,b,x) - y)^2 (where \phi is applied elementwise).
> > The error derivative at (a,b) is therefore:
> >
> > 	G = [ (max{0, ax} - max{0, cx} ) * Ind{ax>0} x, (max{0, bx} - max{0, dx}) * Ind{bx>0} x ]
> >
> > The Frobenius inner product is then given by:
> >
> >         <G, (a,b) - (c, d) >_F = (a-c)(max{0, ax}- max{0, cx}) *Ind{ax>0}x +  (b-d)(max{0, bx}- max{0, dx})*Ind{bx>0}x
> >
> > This is always >= 0 (each term in the sum is >= 0).
> >
> > Another way of looking at a single multi-output layer is that each output and the weights with which it is associated is, in fact, our original GLM problem which we have shown to be SLQC in Theorem 3.4. The separate values of  <G, w_i -v_i > are >= 0 and therefore the sum is also >=0.

---

### Decision · Program_Chairs · 2018-01-29
**ICLR 2018 Conference Acceptance Decision**

**Decision:**

Reject

**Comment:**

Pros:
+ Interesting alternative algorithm for training autoencoders

Cons:
- Not a lot of practical value because DANTE does not outperform SGD in terms of time or classification performance using autoencoder features.

This is an interesting and well-written paper that doesn't quite meet the threshold for ICLR acceptance. If the authors can find use cases where DANTE has demonstrable advantages over competing training algorithms, I expect the paper would be accepted.